# AesExpert: Towards Multi-modality Foundation Model for Image Aesthetics Perception

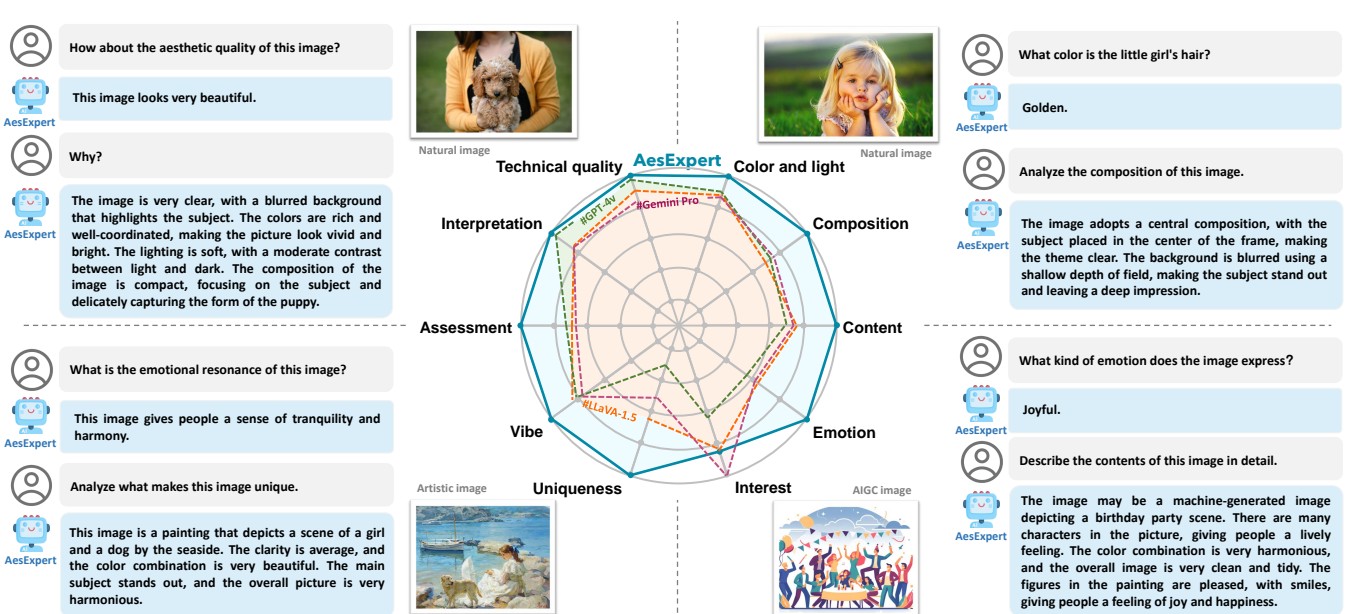

**Figure 1: Performance of the proposed AesExpert on various aesthetic perception dimensions, in comparison with the most advanced GPT-4V and Gemini-Pro-Vision, as well as the open-sourced LLaVA-1.5-13B. (Our AesExpert-Chatbot will be publicly available, providing a more interactive experience in actual application scenarios.)**

## ABSTRACT

The highly abstract nature of image aesthetics perception (IAP) poses significant challenge for current multimodal large language models (MLLMs). The lack of human-annotated multi-modality aesthetic data further exacerbates this dilemma, resulting in MLLMs falling short of aesthetics perception capabilities. To address the above challenge, we first introduce a comprehensively annotated **Aes**thetic **M**ulti-**M**odality **I**nstruction **T**uning (**AesMMIT**) dataset, which serves as the footstone for building multi-modality aesthetics foundation models. Specifically, to align MLLMs with human aesthetics perception, we construct a corpus-rich aesthetic critique database with 21,904 diverse-sourced images and 88K human natural language feedbacks, which are collected via progressive questions, ranging from coarse-grained aesthetic grades to fine-grained aesthetic descriptions. To ensure that MLLMs can handle diverse queries, we further prompt GPT to refine the aesthetic critiques and assemble the large-scale aesthetic instruction tuning dataset, *i.e.* **AesMMIT**, which consists of 409K multi-typed instructions to activate stronger aesthetic capabilities. Based on the AesMMIT database, we fine-tune the open-sourced general foundation models, achieving multi-modality **Aes**thetic **Expert** models, dubbed **AesExpert**. Extensive experiments demonstrate that the proposed AesExpert models deliver significantly better aesthetic perception performances than the state-of-the-art MLLMs, including the most advanced GPT-4V and Gemini-Pro-Vision. The dataset, code and models will be made publicly available.

## CCS CONCEPTS

• **Computing methodologies → Image representations**.

## KEYWORDS

Image aesthetics perception, multi-modality foundation model, natural language feedback, aesthetic critique, instruction tuning.

## 1 INTRODUCTION

Multimodal large language models (MLLMs) have attracted significant attention in the research community [4]. These foundation models, like GPT-4V [49] and LLaVA [34], have demonstrated remarkable progress in serving as general-purpose visual assistants,

### In-figure dialogue text

How about the aesthetic quality of this image?

AesExpert: This image looks very beautiful.

Why?

AesExpert: The image is very clear, with a blurred background that highlights the subject. The colors are rich and well-coordinated, making the picture look vivid and bright. The lighting is soft, with a moderate contrast between light and dark. The composition of the image is compact, focusing on the subject and delicately capturing the form of the puppy.

Natural image

What is the emotional resonance of this image?

AesExpert: This image gives people a sense of tranquility and harmony.

Analyze what makes this image unique.

AesExpert: This image is a painting that depicts a scene of a girl and a dog by the seaside. The clarity is average, and the color combination is very beautiful. The main subject stands out, and the overall picture is very harmonious.

Artistic image

What color is the little girl's hair?

AesExpert: Golden.

Analyze the composition of this image.

AesExpert: The image adopts a central composition, with the subject placed in the center of the frame, making the theme clear. The background is blurred using a shallow depth of field, making the subject stand out and leaving a deep impression.

Natural image

What kind of emotion does the image express?

AesExpert: Joyful.

Describe the contents of this image in detail.

AesExpert: The image may be a machine-generated image depicting a birthday party scene. There are many characters in the picture, giving people a lively feeling. The color combination is very harmonious, and the overall image is very clean and tidy. The figures in the painting are pleased, with smiles, giving people a feeling of joy and happiness.

AIGC image

AesExpert · GPT-4V · Gemini Pro · LLaVA-1.5

Technical quality · Color and light · Composition · Content · Emotion · Interest · Uniqueness · Vibe · Assessment · Interpretation

capable of interacting and collaborating with users [46, 47]. Concurrently, their outstanding cross-modality capacities further promote a paradigm shift in the computer vision domain, with researchers seeking to transcend the limitations of traditional task-specific approaches and develop multi-modality foundation models that generalize effectively across various visual tasks [28, 34, 45]. Despite the advancements achieved, experiments on current MLLMs reveal obvious limitations in the **highly-abstract image aesthetics perception** task [20], which covers not only the extensively-studied image aesthetics assessment (IAA) [50], but also fine-grained aesthetic attribute evaluation (e.g., *color, light,* and *composition*), aesthetic emotion analysis, and image aesthetics caption [15, 23, 40]. The aesthetic perception abilities of MLLMs are crucial for a wide range of practical applications, such as smart photography, album management, photo recommendation, and image enhancement [19, 30]. **Consequently, it is urgent to build a unified foundation model that possesses general abilities across these aesthetic tasks and can precisely respond to open-ended human queries on image aesthetics perception.**

Due to the inherently data-hungry nature, current MLLMs rely on vast amounts of multi-modality instruction-following data to meet general-purpose visual and language understanding [3]. Therefore, researchers have constructed numerous instruction fine-tuning datasets, such as COCO-VQA [2], Science QA [35] and LLaVA-Instruct-150K [33]. However, existing instruction fine-tuning datasets are mainly engineered to enhance the general capacities of MLLMs, *e.g.* visual question answering [2], image captioning [16], object segmentation [26] and content understanding [4]. **A notable gap in these datasets is the inadequate focus on visual aesthetics.** To address the dilemma, we construct a comprehensively annotated **Aes**thetic **M**ulti-**M**odality **I**nstruction **T**uning (**AesMMIT**) dataset, based on which we further fine-tune the open-sourced general foundation models, achieving multi-modality **Aes**thetic **Expert** models, dubbed **AesExpert**, which delivers significantly better aesthetic perception performances than the state-of-the-art MLLMs. Intuitive comparisons and examples are shown in Figure 1. Specifically, this work encompasses three stages:

*Stage 1: Collecting human aesthetic feedback from subjective experiments.* To bridge the gap between MLLMs and human aesthetics perception, we invite human subjects to provide direct feedback on aesthetic perception and understanding via progressive questions, including three parts: 1) The **coarse-grained aesthetic evaluation** (*e.g. This image looks quite beautiful/unattractive.* ). 2) The **fine-grained reasoning and explanation** based on elemental aesthetic attributes (*e.g. clarity, color, light, image object* and *composition, etc.*). 3) The **finer-grained description** on aesthetic feeling (*e.g. novel shooting view, interesting content* and *expressed emotions*). With the three parts, the collected human feedbacks, denoted as **AesFeedback**, can capture the basic aesthetic perceptions and the evaluation reasoning process. The AesFeedback dataset contains **88K human feedbacks** on **21,904 multi-sourced images**.

*Stage 2: Refining feedback with GPT for instruction-following data.* The constructed **AesFeedback** dataset plays a crucial role in fine-tuning MLLMs for aesthetic instructions. However, to fully harness aesthetic capabilities, the dataset should also encompass an aesthetic question-answering component. To obtain rich question-answer pairs, inspired by the existing works (*e.g.* COCO-VQA [32]

and ShareGPT [6]), we leverage GPT to transform human feedback into instruction-following formats, which include both open-end and multiple-choice question-answer pairs. To ensure that MLLMs can handle diverse queries, the instruction-following pairs cover diversified aesthetic perception dimensions (*e.g. quality, attribute, emotion, interpretation, enhancement*, and *context reasoning*), and commonly-used question types (*e.g. Yes-or-No, What, How, Why*, and other *open-ended* questions). Through the above operations, we obtain the final **AesMMIT** dataset, which consists of **409K multi-typed instructions** to activate stronger aesthetic capabilities.

*Stage 3: Building multi-modality aesthetics foundation model based on AesMMIT.* We introduce the instruction fine-tuning to improve the open-source MLLMs [33, 52] based on the AesMMIT dataset, which not only enables the models to retain their original general knowledge but also facilitates the aesthetics perception capabilities, obtaining the multi-modality **Aes**thetic **Expert** models, dubbed **AesExpert**.

The contributions of this study are summarized as follows:

• **Aesthetic instruction-following dataset.** We construct a corpus-rich aesthetic critique database with 21,904 diverse-sourced images and 88K human natural language feedbacks to align MLLMs with human aesthetics perception. Further, we prompt GPT to refine the human aesthetic critiques and assemble the large-scale aesthetic instruction tuning dataset (AesMMIT) to ensure that MLLMs can handle diverse queries, which consists of 409K instructions covering multiple aesthetic perception dimensions, to activate stronger aesthetic capabilities of MLLMs.

• **AesExpert model.** We propose multi-modality aesthetic expert models with the aid of the proposed AesMMIT dataset via instruction fine-tuning. Extensive experiments demonstrate that the proposed AesExpert models deliver significantly better aesthetic perception performances than the state-of-the-art MLLMs, including the most advanced GPT-4V and Gemini-Pro.

• **Open-source.** We release the following sources to the community: (1) the constructed aesthetic multi-modality instruction tuning dataset; (2) the proposed AesExpert models including codes and checkpoints; (3) the visual AesExpert-Chatbot demo. We believe this work would shed light on building more advanced MLLMs with comprehensive aesthetic capabilities.

## 2 RELATED WORK

### 2.1 Multi-modality Foundation Models

Recently, large language models (LLMs), such as GPT-3 [38], Flan-T5 [7] and LLaMA [41], have made remarkable progress in pure-textual tasks. The success of LLMs has also driven the research on vision-language interaction, resulting in the development of various multi-modality foundation models, *e.g.* LLaVA [34], MiniGPT-4 [58], mPLUG-Owl [51], Otter [28] and Instruct-BLIP [8]. These models typically contain a pre-trained visual encoder for image processing, an LLM for interpreting instructions and generating responses, and a cross-modality module to align the vision encoder with the LLM [3]. Despite their impressive performance in general-purpose visual tasks, their performance in handling image aesthetics perception remains underexplored. Therefore, **this work is dedicated to building multi-modality aesthetics large language model.**

**Experiment Preparation**

Scene balance    Quality balance    Diverse sources

Natural      Artistic      AIGC

**(a) 21K diverse-sourced images**

Age
Gender   Personality
Education   Health

Art Practitioners
Researchers
Others

• Evaluation
• Attributes
• Emotion
• Suggestions

Aesthetic knowledge training

**(b) 48 human subjects**

**Collecting Human Feedback**

Image    stimulate    Human

**Aesthetic Evaluation:**
This image does **not look beautiful.**

**Reason Explanation:**
The image has **low clarity.** There is an **overexposure** phenomenon, and the overall **brightness is too high.** The **composition** makes it difficult to distinguish the subject. The nearby weeds are disorderly, and the **distant building is also dilapidated, lacking aesthetic appeal.** If the perspective is brought closer and the content of the weeds at the bottom is cropped, the **composition of this image will improve.**

**Emotion Description:**
This image gives **a sense of uneasiness,** the cluttered content makes one feel annoyed and disoriented.

**(c) 88K human natural language feedbacks**

**Generating Instruction-following Data**

Feedbacks    prompt    GPT-4

**Aesthetic Question Answering:**

**Q:** **How** is the aesthetic quality of this image?
**A:** Low.

**Q:** **What** factors influence the aesthetic experience of this image?
**A:** The composition and overexposure.

**Q:** **Is** this image clear?
A: No.

**Q:** **What** aesthetic emotion can this image convey?
**A:** The cluttered content of this image conveys a sense of uneasiness.

**Q:** **How** can the aesthetic quality of this image be **improved**?
**A:** The perspective should be brought closer and the content of the weeds at the bottom should be cropped.

**(d) 409K multi-typed instructions**

**Figure 2: The proposed dataset construction pipeline. First, we filter 21K diverse-sourced images based on scene, quality and source, and invite 48 human subjects who met the comprehensive criteria. Then, we collect 88K human feedbacks on image aesthetics perception. Finally, human feedbacks are converted into 409K instruction-following pairs (the AesMMIT dataset), which are used for aesthetic instruction tuning.**

## 2.2 Multimodal Instruction-following Datasets.

Instruction tuning is an optimization method for multi-modality foundation models, aiming to improve their ability to perform specific tasks [34]. With the development of MLLMs, researchers often employ ChatGPT [38] or GPT-4 [49] to generate diverse and expansive instruction-following data. For example, Zhu *et al.* [58] employed GPT-3.5 to generate and improve detailed captions for high-quality instruction-following data (MiniGPT-4), which are mainly designed for general visual tasks. Liu *et al.* [33, 34] proposed the multimodal instruction-following dataset (LLaVA-Instruct-150K) based on the existing COCO [32] bounding box and caption dataset using GPT-4. In InstructBLIP [8], the authors transformed 13 vision-language tasks (e.g. OCR-VQA [36]) into the instruction-following format for instruction tuning. Chen *et al.* [6] proposed a large-scale image-text dataset featuring 100K highly descriptive captions generated by GPT-4V and 1.2M high-quality captions generated by the proposed caption model, named ShareGPT4V. A more comprehensive survey can be found in [55].

The existing instruction tuning datasets are mainly constructed for general visual tasks, and the lack of image aesthetic data limits the aesthetic perception ability of MLLMs to a large extent. Further, the existing studies suggested that **hallucination** has become one of the most urgent problems for current MLLMs [13, 18], therefore, using MLLM to directly generate instruction fine-tuning data for aesthetic tasks may further exacerbate this situation. To bridge this gap, this paper presents a corpus-rich aesthetic critique database by collecting human natural language feedback, based on which we establish a comprehensively annotated aesthetic multi-modality

instruction tuning dataset, *i.e.* AesMMIT. Further, we propose multi-modality aesthetic expert models based on aesthetic instruction fine-tuning, achieving significantly better aesthetic perception performances.

## 3 DATASET CONSTRUCTION

In this section, we provide a detailed expatiation of the process involved in building the AesMMIT dataset, which is illustrated in Figure 2. Specifically, subsection 3.1 describes the preparation of the subjective experiment including the collection of images and the recruitment of participants. Subsection 3.2 elaborates how we conduct the subjective experiment to obtain the 88K human feedbacks on 21,904 multi-sourced images. Subsection 3.3 explains how we prompt GPT-4 to refine the aesthetic critiques and obtain the 409K instruction-following data covering multiple aesthetic perception dimensions.

## 3.1 Experiment Preparation

**Image Collection:** To guarantee the diversity of image types, we first collect a large number of images from various sources, including *natural images* (NIs), *artistic images* (AIs) and *artificial intelligence-generated images* (AGIs), as shown in Figure 2(a). Then, we use a well-trained scene classification model [39] to automatically predict the scene label for each image, based on which we sampled these images to maintain scene diversity. To reduce the long-tail distribution of randomly sampled images [50], we add 1239 high-aesthetic images from LITE [42] and Impressions [25] datasets, and 1944 low-quality images from SPAQ [11] and KonIQ-10K [17]

**Table 1: Overview of the image source datasets.**

| Type | Dataset | Sampled Size |
|---|---|---|
| NIs | AADB [24] | 1096 |
| | PARA [48] | 3694 |
| | TAD66K [14] | 4003 |
| | LITE [42] | 1087 |
| | Impressions [25] | 152 |
| | SPAQ [11] | 1153 |
| | KonIQ-10K [17] | 791 |
| AIs | BAID [53] | 2970 |
| | CAD [27] | 59 |
| | ArtEmis [1] | 1957 |
| AGIs | DiffusionDB [43] | 4228 |
| | AGIQA-3K [56] | 570 |
| | AGIQA-1K [29] | 144 |

datasets. Finally, a total of 21,904 images are collected, which are further fed to human subjects to collect aesthetic feedbacks based on subjective experiments. Detailed image source of the AesMMIT dataset is summarized in Table 1.

**Subject Selection:** To ensure the completeness of annotation and the diversity of corpus, we recruit subjects by considering five different perspectives, including *age, gender, education, health status* and *personality*, as shown in Figure 2(b). Specifically, we ensure that each subject is in good health and passes the Ishihara color blindness test [48]. To maintain the quality of annotation, all subjects need to have a high school degree or above, and photography experience. Following the existing works [44, 48], to ensure the diversity of annotation and language, we further consider age distribution, gender balance, and personality diversity [31]. Finally, considering the difficulty of labeling artistic and artificial intelligence-generated images, we specially invited 6 art practitioners and 9 researchers in the field of image aesthetics evaluation to join the subjective experiment and focus on labeling these two kinds of images. In addition, we also organize two aesthetic knowledge training sessions before the experiment covering *evaluation, attribute, emotion* and *suggestion*, aiming to provide them with a more comprehensive understanding of image aesthetics annotation. Finally, based on the principle of voluntary participation [48], we invite 48 eligible human subjects to ensure the validity and reliability of annotations.

## 3.2 Collecting Human Aesthetic Feedback

To enhance the understanding and interpretation capabilities of MLLMs in terms of aesthetics perception, We invite all 48 subjects to contribute their insights and descriptions on image aesthetics through progressive questions, as illustrated in Figure 2(c). This process includes three parts for each test image from coarse to fine. A screenshot of the annotation interface is shown in Figure 3.

1) **Coarse-grained aesthetic evaluation.** The focus of this question is to collect simple aesthetic grade judgments on images. Participants could express their overall impression of image aesthetics using simple sentences such as *good-looking, beautiful, average* or *bad-looking, etc.* Referring to common settings in the field of image quality assessment [9, 24], we recommend that experimental

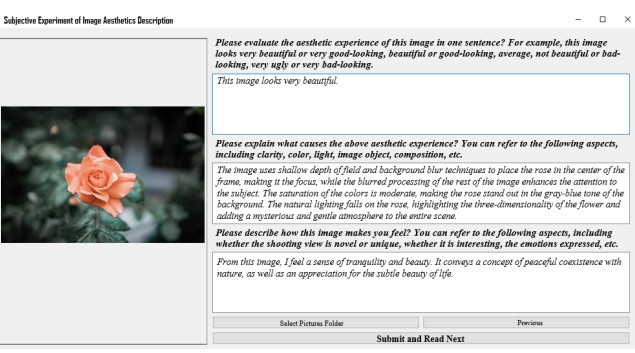

**Figure 3: The screenshot of the annotation interface for the subjective experiment.**

participants establish a uniform measure of five aesthetic grades and guarantee language diversity.

2) **Fine-grained reasoning and explanation.** On the basis of simple aesthetic grade judgments, this question is designed to mine the detailed reasons that influence human aesthetic judgments. Participants need to explain what specific aspects of the image enhance or reduce the aesthetic appeal, such as *clarity, color, light, image object* and *composition*. In addition, we encourage participants to provide suggestions for the improvement of low-aesthetic images.

3) **Finer-grained description on aesthetic feeling.** This question breaks through the description of the inherent aesthetic attributes and aims to explore the impact of human emotions on aesthetic perception when viewing images. Participants need to describe some of the more abstract emotional factors in the image, **e.g.** *novel shooting view, interesting content, expressed emotions, etc.*. Furthermore, we encourage participants to analyze the impact of these factors on image aesthetics.

The setting of the subjective experiment refers to the ITU-R Recommendation BT.500-13 standard [22]. By systematically collecting feedback from the above three parts, we construct the corpus-rich aesthetic critique database, called **AesFeedback**, which contains **88K human feedbacks** on **21,904 multi-sourced images**. Since these annotations are all provided by human subjects, AesFeedback can capture the characteristics of basic aesthetic perceptions and the evaluation reasoning process, providing a valuable resource for improving the aesthetic capabilities of MLLM.

## 3.3 Generating Instruction-following Data

While the corpus-rich aesthetic critiques in the **AesFeedback** dataset can provide rich knowledge for aesthetic instruction tuning of MLLMs, we further design more instruction-following data to allow MLLMs to respond to a variety of human queries, achieving stronger aesthetic capabilities. Similar to existing works [6, 45], we leverage GPT-4 to transform human feedback into instruction-following formats, as illustrated in Figure 2(d). Through this process, the proposed AesMMIT dataset includes 409K instruction-response pairs, with its details as follows.

**Aesthetic Description:** Similar to aesthetic caption [40], the ability of image aesthetic description is crucial for MLLMs. As shown in Figure 2(c), the collected AesFeedback dataset contains

direct and comprehensive human natural language critiques on image aesthetics. Furthermore, these critiques provide aesthetic interpretation from overall aesthetic grades to fine-grained aesthetic attributes (*e.g. clarity, color, light*, and *image object*), which can activate the preliminary aesthetic interpretation abilities of MLLMs (see Figure 5). Therefore, we directly use the questions in the AesFeedback dataset as instructions and human critiques as responses, based on which we obtain the 88K instruction-following pairs of the proposed AesMMIT dataset.

**Aesthetic Question Answering:** In addition to directly adopting the AesFeedback as aesthetic instruction-following data, we further propose a GPT-assist approach to refine the aesthetic critiques and assemble a larger-scale aesthetic question-answering subset (AesVQA). Inspired by COCO-VQA [32] and ShareGPT [6], we prompt GPT-4 to generate diverse-style open-ended questions and provide corresponding brief answers, where both the questions and answers are based on human aesthetic descriptions. To ensure that MLLMs can handle diverse queries during the interaction, the questions are generated to cover commonly used question types. Specifically, **Yes-or-No-style** questions are straightforward queries that demand a simple *yes* or *no* as an answer, which are mainly used to provide a clear, binary response. **What-style** questions are leveraged to measure more comprehensive and complex aesthetic perception (*e.g.* various aesthetic attributes). **How-style** questions are used to ask more details about aesthetic emotion, aesthetic attributes, and improvement suggestions; **Why-style** questions are employed to explore the foundational aspects of what makes images aesthetically appealing or unappealing, revealing the rationale of beauty and visual appeal. To ensure the diversity of instructions, following the existing works [33, 35], besides the direct answers, we also generate several distracting answers for the questions and convert them into an additional multi-choice question format [47]. Through the above process, we obtain the final **AesMMIT** dataset, which consists of **409K multi-typed instructions**, aiming to activate stronger aesthetic capabilities. Figure 4 shows the frequently occurring words in the proposed AesMMIT dataset. It can be observed that most nouns, adjectives and adverbs are related to image aesthetic description, which is quite different from common semantic-based tasks. More details are provided in the Supplementary.

## 4 MODEL FINE-TUNING

To verify the efficacy of the proposed AesMMIT dataset, we introduce instruction fine-tuning to improve the open-source MLLMs to enhance the aesthetic perception capabilities, obtaining multi-modality **Aes**thetic **Expert** models.

### 4.1 Model Architecture

The proposed AesExpert model follows the design of LLaVA-1.5 [33], which includes three components: (1) A visual model based on the CLIP-ViT-L14 [39] with an input size of 336×336, which converts the input image into 576 tokens. (2) A visual-language projector based on two-layer multi-layer perception (MLP), which is employed to connect the visual modality and language modality. (3) A language model based on the open-source Vicuna-v1.5 [57], which is used to interpret instructions and generate responses. In this work, we

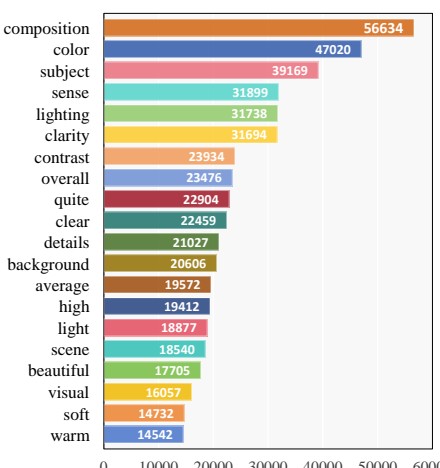

**Figure 4: Statistics of top-20 frequently-used words in AesMMIT database.**

**Table 2: Structures of the multi-modality foundation models for aesthetic instruction tuning.**

| Model | Visual Model | V→L | Language Model |
|---|---|---|---|
| LLaVA-1.5-7B | CLIP-ViT-L14 | MLP | Vicuna-v1.5-7B |
| LLaVA-1.5-13B | CLIP-ViT-L14 | MLP | Vicuna-v1.5-13B |
| mPLUG-Owl2 | CLIP-ViT-L14 | Abstractor | LLaMA2-7B |

build the AesExpert models based on two variants of LLaVA-1.5, including LLaVA-1.5-7B and LLaVA-1.5-13B. In addition, we also introduce the mPLUG-Owl2 [52] as the backbone to validate the aesthetic perception performance based on the AesMMIT dataset. The structures for these multi-modality foundation models are summarized in Table 2.

### 4.2 Supervised Fine-Tuning

In general, the training of open-source MLLMs [33, 34, 58] includes two stages: aligning the representation space of the visual backbone and the LLM with million-scale web data [35], and visual instruction tuning with a combination of multi-modality datasets [6, 8]. Considering that our purpose is to improve the aesthetic perception ability of the current MLLMs, we directly use the proposed AesMMIT dataset to perform supervised instruction fine-tuning on the models pre-trained on general-purpose visual tasks [33]. Following existing works [5, 45], supervised instruction fine-tuning not only enables the models to retain their original general knowledge but also facilitates the aesthetics perception capabilities. In this work, to enhance the training efficiency and compare fairly, we freeze the vision model and focus on fine-tuning the projector and the language model. Based on supervised instruction fine-tuning, we implemented three different versions of multi-modality aesthetic expert models.

**Table 3: Performance comparisons of the proposed AesExpert with existing MLLMs. AesA: Aesthetic Assessment, AesI: Aesthetic Interpretation, T. Q.: Technical quality, C. L.: Color and light, Comp.: Composition, Cont.: Content, Emot.: Emotion, Inte.: Interest, Uniq.: Uniqueness, L-13B: LLaVA-1.5-13B, Owl2: mPLUG-Owl2, L-7B: LLaVA-1.5-7B. The overall score represents the average of the four dimension scores.**

| Model | Aesthetic Perception | | | | | Aesthetic Empathy | | | | | AesA | AesI | Overall | Rank |
|---|---|---|---|---|---|---|---|---|---|---|---|---|---|---|
| | T. Q. | C. L. | Comp. | Cont. | **Score** | Emot. | Inte. | Uniq. | Vibe | **Score** | **Score** | **Score** | | |
| **AesExpert (L-13B)** | 70.83% | **80.63%** | **81.75%** | **76.60%** | **79.54%** | **82.94%** | 80.65% | **93.10%** | **88.56%** | **84.89%** | **59.57%** | **1.340** | **89.50%** | **1** |
| AesExpert (Owl2) | 70.19% | 79.47% | 79.56% | 71.28% | 77.64% | 80.66% | 83.87% | 89.66% | 86.31% | 82.68% | 56.32% | 1.336 | 87.56% | 2 |
| AesExpert (L-7B) | **72.44%** | 79.38% | 80.66% | 74.47% | 78.57% | 82.05% | 83.87% | 86.21% | 87.81% | 84.04% | 53.25% | 1.317 | 86.89% | 3 |
| GPT-4V | 69.02% | 74.66% | 71.72% | 65.57% | 72.08% | 65.06% | 72.41% | 62.07% | 80.15% | 70.16% | 50.86% | 1.301 | 80.80% | 4 |
| ShareGPT4V | 62.18% | 71.90% | 69.29% | 64.89% | 69.18% | 66.48% | 80.65% | 68.97% | 78.72% | 70.75% | 47.82% | 1.296 | 79.34% | 5 |
| LLaVA-1.5-13B | 67.63% | 74.65% | 70.09% | 68.44% | 71.61% | 67.15% | 80.65% | 75.86% | 81.18% | 72.07% | 49.82% | 1.222 | 78.93% | 6 |
| Gemini Pro Vision | 65.08% | 74.57% | 72.24% | 67.97% | 71.99% | 66.87% | **87.50%** | 70.00% | 79.09% | 71.37% | 49.38% | 1.222 | 78.74% | 7 |
| Q-Instruct | 66.03% | 74.48% | 73.68% | 68.09% | 72.61% | 68.64% | 83.86% | 75.86% | 80.00% | 72.68% | 52.86% | 1.020 | 75.04% | 8 |
| GLM | 55.77% | 54.61% | 51.25% | 48.94% | 52.96% | 53.13% | 70.97% | 44.83% | 55.29% | 53.96% | 37.79% | 0.932 | 59.48% | 9 |
| MiniGPT-4 | 39.42% | 41.31% | 42.67% | 44.33% | 41.93% | 39.78% | 38.71% | 24.14% | 39.04% | 39.35% | 38.57% | 0.999 | 54.94% | 10 |
| mPLUG-Owl2 | 60.90% | 70.57% | 68.30% | 62.77% | 67.89% | 65.60% | 77.42% | 65.52% | 78.07% | 69.89% | 50.57% | 1.182 | 47.38% | 11 |
| LLaVA-1.5-7B | 53.85% | 70.16% | 67.40% | 59.93% | 66.32% | 62.49% | 80.65% | 75.85% | 78.93% | 68.32% | 45.46% | 1.157 | 45.31% | 12 |
| Qwen-VL | 54.81% | 66.25% | 62.91% | 60.64% | 63.21% | 58.67% | 83.87% | 72.41% | 73.90% | 64.18% | 46.25% | 1.192 | 43.71% | 13 |
| LLaVA | 46.79% | 63.59% | 65.30% | 64.54% | 62.43% | 58.61% | 80.63% | 65.52% | 75.83% | 64.68% | 45.96% | 1.125 | 43.55% | 14 |
| TinyGPT-V | 21.79% | 24.52% | 22.13% | 28.01% | 23.71% | 30.36% | 29.03% | 31.03% | 35.40% | 32.04% | 43.57% | 0.701 | 42.36% | 15 |
| InstructBLIP | 37.82% | 55.36% | 55.43% | 57.09% | 54.29% | 49.64% | 58.06% | 51.72% | 61.50% | 53.89% | 46.54% | 1.126 | 38.96% | 16 |
| Otter | 35.90% | 54.28% | 51.65% | 51.06% | 50.96% | 48.42% | 70.97% | 51.72% | 63.21% | 53.64% | 44.86% | 1.027 | 37.62% | 17 |
| IDEFICS-Instruct | 37.50% | 52.87% | 52.84% | 51.06% | 50.82% | 43.93% | 64.52% | 62.07% | 64.06% | 50.82% | 45.00% | 1.180 | 36.96% | 18 |
| MiniGPT-v2 | 56.73% | 56.44% | 51.74% | 50.00% | 54.18% | 52.52% | 58.06% | 44.83% | 58.07% | 54.36% | 31.11% | 1.003 | 35.16% | 19 |

# 5 EXPERIMENTS

## 5.1 Implementation Details

**Training setting**: In this work, we finetune three pre-trained MLLMs based on the constructed AesMMIT dataset in full schedule mode, including LLaVA-1.5-7B (Vicuna-v1.5-7B) [33], LLaVA-1.5-13B (Vicuna-v1.5-13B) [33] and mPLUG-Owl2 (LLaMA-2-7B) [52]. In implementation, to ensure fairness, we follow the default hyperparameters provided by the original models. All models are trained based on 8 NVIDIA Tesla A100 80G GPUs, and the evaluation experiments are conducted on 2 NVIDIA RTX 4090 24G GPUs.

**Benchmark**: To verify the performance of our proposed AesExpert models on image aesthetics perception, extensive experiments and comparisons are conducted on the AesBench [20], which is a well-designed benchmark for MLLMs on aesthetics perception evaluation. Specifically, AesBench contains 2,800 images and four evaluation criteria designed from four dimensions, including (1) **Aesthetic Perception** (*AesP*) focuses on the ability of MLLMs to recognize and understand aesthetic attributes. (2) **Aesthetic Empathy** (*AesE*) evaluates the ability of MLLMs to resonate with the emotional aspects conveyed through aesthetic expressions like humans. (3) **Aesthetic Assessment** (*AesA*) evaluates the ability of MLLMs to judge aesthetic grades and predict quality scores based on the language description. (4) **Aesthetic Interpretation** (*AesI*) involves the ability of MLLMs to interpret and analyze the reasons for aesthetic quality. Each dimension contains 2,800 questions and correct answers. For the first three dimensions, the accuracy of the answers is used to measure the performance of the model, while the AesI is evaluated based on GPT scoring.

## 5.2 Performance Comparison

In this section, we compare the performance of our proposed AesExpert models (three versions) with 16 state-of-the-art MLLMs, including the popular GPT-4V [49] and Gemini Pro Vision [12], as well as 13 state-of-the-art variants with open sources, *i.e.* LLaVA (LLaMA-2-Chat-7B) [34], LLaVA-1.5-7B (Vicuna-v1.5-7B) [33], LLaVA-1.5-13B (Vicuna-v1.5-13B) [33], ShareGPT4V (Vicuna-v1.5-7B) [6], Q-Instruct (LLaVA-v1.5-7B) [45], mPLUG-Owl2 (LLaMA-2-7B) [52], InstructBLIP (Vicuna-7B) [8], MiniGPT-4 (Vicuna-7B) [58], MiniGPT-v2 (LLaMA-2-Chat-7B) [5], IDEFICS-Instruct (LLaMA-7B) [21], GLM (ChatGLM-6B) [10], Otter (MPT-7B) [28], TinyGPT-V (Phi-2) [54] and Qwen-VL (QWen-7B) [3]. Detailed information about these models can be found in [55]. The results are listed in Table 3.

From Table 3, we can find that the three versions of AesExpert achieve the top three results. Among them, AesExpert based on LLaVA-1.5-13B achieves the best performance, which is significantly better than the most advanced GPT-4V. For the existing open-source models, ShareGPT4V [6] performs best, but lags behind our AesExpert (L-13B) by more than 10%. These experimental results reveal that the proposed AesExpert models have the best aesthetic perception abilities and highlight the advantage of the constructed AesMMIT dataset for improving multi-modality foundation models.

## 5.3 Performance Improvement

In this section, we quantitatively evaluate the aesthetic perception abilities of MLLMs after aesthetic instruction tuning in the four tasks defined by AesBench [20]. The experimental results are summarized in Tables 4, 5, 6 and 7, respectively.

**Table 4: Comparison of the Aesthetic Perception ability between baseline MLLMs and the proposed AesExpert models.**

| MLLM | Perceptual Dimensions | | | | Image Sources | | | Question Types | | | | Overall |
|---|---|---|---|---|---|---|---|---|---|---|---|---|
| | Tec. Qua. | Col. Lig. | Composition | Content | NIs | AIs | AGIs | Yes-No | What | How | Why | |
| BL (mPLUG-Owl2) | 60.90% | 70.57% | 68.30% | 62.77% | 72.23% | 64.71% | 64.10% | 65.59% | 58.64% | 73.02% | 80.73% | 67.89% |
| **AesExpert** | **70.19%** | **79.47%** | **79.56%** | **71.28%** | **79.78%** | **73.32%** | **78.72%** | **72.75%** | **66.43%** | **89.32%** | **89.02%** | **77.64%** |
| Improvement | +9.29% | +8.90% | +11.26% | +8.51% | +7.55% | +8.61% | +14.62% | +7.16% | +7.79% | +16.30% | +8.29% | +9.75% |
| BL (LLaVA-1.5-7b) | 53.85% | 70.16% | 67.40% | 59.93% | 69.10% | 65.71% | 62.37% | 62.36% | 58.92% | 70.71% | 81.22% | 66.32% |
| **AesExpert** | **72.44%** | **79.38%** | **80.66%** | **74.47%** | **81.46%** | **74.69%** | **77.93%** | **70.74%** | **72.52%** | **90.33%** | **88.05%** | **78.57%** |
| Improvement | +18.59% | +9.22% | +13.26% | +14.54% | +12.36% | +8.98% | +15.56% | +8.38% | +13.60% | +19.62% | +6.83% | +12.25% |
| BL (LLaVA-1.5-13b) | 67.63% | 74.65% | 70.09% | 68.44% | 75.36% | 70.32% | 66.76% | 68.72% | 61.19% | 78.21% | 85.37% | 71.61% |
| **AesExpert** | **70.83%** | **80.63%** | **81.75%** | **76.60%** | **81.54%** | **76.93%** | **78.99%** | **71.14%** | **70.40%** | **93.80%** | **91.46%** | **79.54%** |
| Improvement | +3.20% | +5.98% | +11.66% | +8.16% | +6.18% | +6.61% | +12.23% | +2.42% | +9.21% | +15.59% | +6.09% | +7.93% |
| Average Impro. | +10.36% | +8.03% | +12.06% | +10.40% | +8.70% | +8.07% | +14.14% | +5.99% | +10.20% | +17.17% | +7.07% | +9.98% |

**Table 5: Comparison of the Aesthetic Empathy ability between baseline MLLMs and the proposed AesExpert models.**

| MLLM | Empathy Dimensions | | | | Image Sources | | | Question Types | | | | Overall |
|---|---|---|---|---|---|---|---|---|---|---|---|---|
| | Emotion | Interest | Uniqueness | Vibe | NIs | AIs | AGIs | Yes-No | What | How | Why | |
| BL (mPLUG-Owl2) | 65.60% | 77.42% | 65.52% | 78.07% | 71.03% | 71.57% | 66.22% | 68.05% | 64.16% | 70.14% | 83.82% | 69.89% |
| **AesExpert** | **80.66%** | **83.87%** | **89.66%** | **86.31%** | **84.27%** | **80.42%** | **82.45%** | **72.52%** | **80.59%** | **94.29%** | **90.93%** | **82.68%** |
| Improvement | +15.06% | +6.45% | +24.14% | +8.24% | +13.24% | +8.85% | +16.23% | +4.47% | +16.43% | +24.15% | +7.11% | +12.79% |
| BL (LLaVA-1.5-7b) | 62.49% | 80.65% | 75.85% | 78.93% | 69.26% | 69.58% | 65.43% | 62.37% | 64.16% | 71.71% | 84.07% | 68.32% |
| **AesExpert** | **82.05%** | **83.87%** | **86.21%** | **87.81%** | **86.36%** | **82.54%** | **81.78%** | **74.04%** | **83.14%** | **95.29%** | **90.44%** | **84.04%** |
| Improvement | +19.56% | +3.22% | +10.36% | +8.88% | +17.10% | +12.96% | +16.35% | +11.67% | +18.98% | +23.58% | +6.37% | +15.72% |
| BL (LLaVA-1.5-13b) | 67.15% | 80.65% | 75.86% | 81.18% | 72.79% | 74.44% | 68.35% | 70.28% | 65.16% | 73.86% | 85.29% | 72.07% |
| **AesExpert** | **82.94%** | **80.65%** | **93.10%** | **88.56%** | **86.20%** | **84.04%** | **83.64%** | **74.14%** | **84.84%** | **95.57%** | **92.65%** | **84.89%** |
| Improvement | +15.79% | +0% | +17.24% | +7.38% | +13.41% | +9.60% | +15.29% | +3.86% | +19.68% | +21.71% | +7.36% | +12.82% |
| Average Impro. | +16.80% | +3.22% | +17.25% | +8.17% | +14.58% | +10.47% | +15.96% | +6.67% | +18.36% | +23.15% | +6.94% | +13.78% |

**Aesthetic Perception Ability.** From Table 4, we can observe that fine-tuning baseline MLLMs using AesMMIT can significantly improve their image aesthetic perception abilities. Specifically, among the three baseline MLLMs, LLaVA-1.5-7b achieves the most performance improvements (over 12% on the overall score). For the four different perception dimensions, we noticed the most significant improvement for **composition**. The possible reason is that composition is a very important element in human aesthetic perception, which also can be found in Figure 4, indicating that composition is the most frequently occurring word in the AesMMIT dataset. Therefore, the proposed AesExpert model fine-tuned on AesMMIT has excellent composition perception capabilities. In addition, among the three types of images, **artificial intelligence-generated images** obtain the biggest performance improvement. This is mainly because that existing instruction fine-tuning datasets usually contain very few artificial intelligence-generated images, and our AesMMIT dataset makes up for this shortcoming, achieving significant performance improvement. Finally, the biggest performance improvement of the four question types is '*How*'. These findings inspire us to further expand our dataset to cover more perception dimensions and more question types in future studies.

**Aesthetic Empathy Ability.** From Table 5, it is observed that our AesExpert is superior to the baseline models by a large margin (more than 12%), especially on LLaVA-1.5-7B, with performance

**Table 6: Comparison of the Aesthetic Assessment ability between baseline MLLMs and the proposed AesExpert models.**

| MLLM | NIs | AIs | AGIs | Overall |
|---|---|---|---|---|
| BL (mPLUG-Owl2) | 57.78% | 49.50% | 40.83% | 50.57% |
| **AesExpert** | **64.45%** | **51.50%** | **48.01%** | **56.32%** |
| Improvement | +6.67% | +2.00% | +7.18% | 5.75% |
| BL (LLaVA-1.5-7b) | 50.08% | **48.13%** | 34.97% | 45.46% |
| **AesExpert** | **61.40%** | 47.63% | **45.74%** | **53.25%** |
| Improvement | +11.32% | -0.50% | +10.77% | +7.79% |
| BL (LLaVA-1.5-13b) | 56.66% | 49.63% | 38.70% | 49.82% |
| **AesExpert** | **67.82%** | **53.74%** | **52.13%** | **59.57%** |
| Improvement | +11.16% | +4.11% | +13.43% | +9.75% |
| Average Impro. | +9.72% | +1.87% | +10.46% | +7.76% |

improvement up to 15.72%. Moreover, the average improvement on **Uniqueness** is more significant than on other dimensions, indicating that the major concerns for empathy raised by humans in the AesMMIT dataset are related to the uniqueness of perspective. Among the three types of images, **artificial intelligence-generated images** still obtain the biggest performance improvement (more than 15%). In addition, for the four question types,

Table 7: Comparison of the Aesthetic Interpretation ability between baseline MLLMs and the proposed AesExpert models. Rele.: Relevance, Prec.: Precision, Comp.: Completeness.

| MLLM | Rele. | Prec. | Comp. | Overall |
|---|---|---|---|---|
| BL (mPLUG-Owl2) | 1.402 | 1.016 | 1.130 | 1.182 |
| **AesExpert** | 1.406 | 1.431 | 1.171 | 1.336 |
| Improvement | +0.40% | +41.50% | +4.10% | +15.40% |
| BL (LLaVA-1.5-7b) | **1.374** | 0.918 | 1.084 | 1.125 |
| **AesExpert** | 1.379 | **1.399** | **1.171** | **1.317** |
| Improvement | +0.50% | +48.10% | +8.70% | +19.20% |
| BL (LLaVA-1.5-13B) | **1.403** | 1.150 | 1.113 | 1.222 |
| **AesExpert** | 1.412 | **1.409** | **1.198** | **1.340** |
| Improvement | +0.90% | +25.90% | +8.50% | +11.80% |
| Average Impro. | +0.6% | +38.5% | +7.1% | +15.47% |

*HOW* questions achieve the biggest performance improvement. In summary, these results underscore that AesMMIT can significantly improve the aesthetic empathy abilities of MLLMs, and even our AesExpert based on LLaVA-1.5-7B markedly surpasses the current top-performing GPT-4V (refer to Table 3).

**Aesthetic Assessment Ability.** The observations from Table 6 underscore that the aesthetic instruction tuning also notably improves the aesthetic assessment ability of MLLMs, especially on the **artificial intelligence-generated images** with an average improvement of 10.46%. In contrast, the average improvement on *artistic image* (+1.87%) is less significant, implying that, due to the highly abstract nature, the aesthetic assessment of artistic images is still a relatively difficult task. We look forward to better solutions for artistic image in the future.

**Aesthetic Interpretation Ability**. Hallucination has been regarded as one of the critical challenges for MLLMs [13, 18], which imagines incorrect details about an image in visual question answering. To alleviate this problem, the proposed AesMMIT dataset is collected from **human natural language feedback** rather than machine-generated annotations. As can be seen from Table 7, the **Precision** of interpretation has been significantly improved. For three different baseline MLLMs, significant improvements of 41.50%, 48.10% and 25.90% have been achieved improved, respectively. This result proves that the proposed AesMMIT allows the MLLMs to learn the style of human language and enhance the precision of aesthetic descriptions. In addition, from the overall score, the aesthetic instruction tuning based on AesMMIT significantly improves the aesthetic interpretation ability of MLLMs (average performance improvement of 15.47%), especially for LLaVA-1.5-7B, with a performance improvement up to 19.20%. These results demonstrate that the AesMMIT dataset could significantly benefit the existing MLLMs for obtaining enhanced aesthetic interpretation ability.

### 5.4 Comparison of Training Data

To verify the effectiveness and necessity of collecting human feedback for improving the aesthetic perception abilities of MLLMs, we further conduct experiments to compare our dataset with the AVA-Comments dataset [37], the largest multi-modality dataset in the image aesthetics domain, which contains over 250K images with

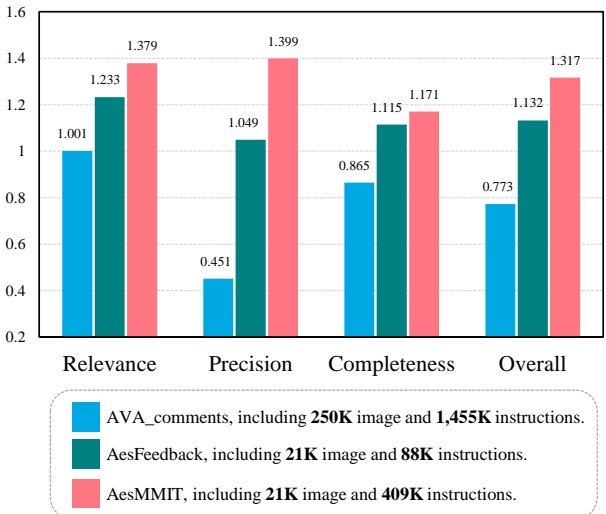

Figure 5: Performance comparison using different datasets.

1,455K comments. Considering that the AVA-Comments dataset only contains aesthetic descriptions, for fairness, we evaluate the aesthetic interpretation ability across different datasets using the same instruction settings. In addition, we use the AesFeedback subset and the whole AesMMIT dataset to fine-tune the model for comparison, respectively. Figure 5 provides the experimental results, where all experiments adopt the same LLaVA-1.5-7B as the baseline MLLM.

It can be observed from Figure 5 that although AVA-Comments contains more images and instructions, it cannot provide effective aesthetic information to MLLM, resulting in poor aesthetic interpretation ability, especially on **Precision**. In contrast, the human feedback we collected (AesFeedback) can achieve pretty good aesthetic interpretation abilities for MLLMs. More importantly, the proposed AesMMIT dataset expanded by GPT can further improve the model performance. These results clearly demonstrate the effectiveness and necessity of the proposed AesMMIT dataset.

## 6 CONCLUSION

In this work, we have made an attempt to exploit the aesthetic perception ability of the multi-modality foundation model. Specifically, we first build a corpus-rich aesthetic critique database via human natural language feedback (**AesFeedback**), based on which we further establish a comprehensively annotated aesthetic multi-modality instruction tuning dataset (**AesMMIT**). In addition, we propose multi-modality aesthetic expert models based on aesthetic instruction fine-tuning, achieving significantly better aesthetic perception performances. We believe this work is a solid step in improving the aesthetic perception ability of MLLM, and we hope that our contribution will encourage the research community to build multi-modality foundation models that can understand highly abstract image aesthetics like humans.

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
