# OpenReview forum: "AesExpert: Towards Multi-modality Foundation Model for Image Aesthetics Perception"
_acmmm.org/ACMMM/2024/Conference — MM2024 Poster_

### Official Review · Reviewer_2oS2 · 2024-05-20

**Rating:** 4
**Confidence:** 4

**Summary:**

This paper explores the problem of using multi-modality foundation models to handle image aesthetic perception task. To this end, the author constructed an aesthetic comment database (AesFeedback) through human natural language feedback, and further established an aesthetic multi-modal instruction tuning dataset based on this. Based on the AesMMIT database, the author fine-tuned the open source general foundation model and implemented a AesExpert model. Experiments demonstrate that the proposed AesExpert model has better aesthetic perception performance than state-of-the-art multi-modal large language models.

**Strengths:**

1.	This study fills the gap of existing instruction tuning datasets for image aesthetic. The authors provide a new research path for multimodal aesthetic perception by collecting large amounts of human natural language feedback and constructing a new AesMMIT dataset.
2.	The paper verifies the performance of the AesExpert model through many experiments. The experimental results show that the AesExpert model is significantly better than the SOTA MLLMs in aesthetic perception performance.

**Limitations:**

1.	It would be beneficial to provide more examples of image-label pairs from the AesMMIT dataset.
2.	In Section 5.4, you mentioned conducting experiments to compare your dataset with the AVA-Comments dataset to verify the effectiveness and necessity of collecting human feedback for improving the aesthetic perception abilities of MLLMs. However, it is worth noting that the AVA-Comments dataset also consists of comments from human feedback. How to prove the above point?
3.	AVA-Captions (Aesthetic Image Captioning From Weakly-Labelled Photographs) dataset contains aesthetic comments that have been screened. What are the results when using this data set?
4.	More details about the AesExpert model, including its architecture and training process, would be helpful. This information would provide a deeper understanding of the model and could aid others in replicating or building upon your work.

**Suitability:**

3

---

### Official Review · Reviewer_Qtwa · 2024-05-25

**Rating:** 5
**Confidence:** 3

**Summary:**

The highly abstract nature of image aesthetics perception (IAP) poses significant challenge for current multimodal large language models (MLLMs). The lack of human-annotated multi-modality aesthetic data further exacerbates this dilemma, resulting in MLLMs falling short of aesthetics perception capabilities.

This work introduces a comprehensively annotated Aesthetic Multi-Modality Instruction Tuning (AesMMIT) dataset. The database contains 21,904 diverse-sourced images and 88K human natural language feedbacks, which are collected via progressive questions, ranging from coarse-grained aesthetic grades to fine-grained aesthetic descriptions. The authors prompt GPT to refine the aesthetic critiques and assemble the large-scale aesthetic instruction tuning dataset, i.e. AesMMIT, which consists of 409K multi-typed instructions to activate stronger aesthetic capabilities. Based on the AesMMIT database, they fine-tune the open-sourced general foundation models, achieving multi-modality Aesthetic Expert models, AesExpert.

**Strengths:**

Extensive experiments demonstrate that the proposed AesExpert models deliver significantly better aesthetic perception performances than the state-of-the-art MLLMs, including the most advanced GPT-4V and Gemini-Pro-Vision.

**Limitations:**

Image aesthetic assessment has always been a very abstract yet highly significant task. This work constructs a large-scale dataset and corresponding evaluation model, which greatly promotes the progress of aesthetic assessment.

**Suitability:**

3

---

### Official Review · Reviewer_n3hH · 2024-05-27

**Rating:** 4
**Confidence:** 2

**Summary:**

This paper presents a human-annotated multi-modality aesthetic dataset named AesMMIT, which consists of 409K multi-typed instructions. Based on the AesMMIT database, we fine-tune the open-sourced general foundation models, achieving multi-modality Aesthetic Expert models, dubbed AesExpert. Extensive experiments demonstrate the efficacy of AesMMIT and AesExpert.

**Strengths:**

**Strengths**

Good writing, well motivated with an interesting introduction/motivation, well presented methods section with very good illustrations.

**Limitations:**

**Limitations**

I have two main concerns regarding this paper:

* I am aware of several existing datasets that use MLLMs, i.e., GPT-4v related to image aesthetics or image quality, like VisionPrefer and Q-Bench. I believe the author should conduct comparative ablation studies on these datasets (if author can obtain them).

* I am somewhat concerned that if incomplete information appears in human feedback, it might prevent GPT-4 from correctly generating the instructed data.

**Suitability:**

2

---

### Meta-Review · Area_Chair_VhD9 · 2024-07-01

**Recommendation:** Accept (Poster)
**Confidence:** 4

**Metareview:**

The paper proposes a method for image aesthetics perception using MLLMs.

The reviewers mention (+) improving aesthetic perception performance over SOTA MLLM methods, (+) the large-scale dataset and corresponding evaluation model improving the state of this subjective task, (+) many experiments showing the capabilities of the model.

They are however concerned about (-) comparisons to the AVA Captions dataset, (-) some missing data from the paper, (-) missing comparison to VisionPrefer and Q-Bench.

The reviews were all tending positively, with initially two borderline accepts and one weak accept. After the rebuttal, it resulted in one borderline accept and two weak accepts. One reviewer judged for "poster", another mentioned that while it has some remaining issues, it has "valuable insights to the multimedia field".

I believe overall this work should be accepted and I'll recommend doing so as poster.